# Subgrid Variational Optimized Optical Flow Estimation Algorithm for Image Velocimetry

**DOI:** 10.3390/s23010437

**Published:** 2022-12-30

**Authors:** Haoxuan Xu, Jianping Wang, Ya Zhang, Guo Zhang, Zhaolong Xiong

**Affiliations:** 1Faculty of Information Engineering and Automation, Kunming University of Science and Technology, Kunming 650000, China; 2Nanjing Institute of Water Resources and Hydrology Automation, Ministry of Water Resources, Nanjing 210000, China

**Keywords:** optical flow, sub-grid scale, large eddy simulation, turbulence, open channel

## Abstract

The variational optical flow model is used in this work to investigate a subgrid-scale optimization approach for modeling complex fluid flows in image sequences and estimating their two-dimensional velocity fields. To solve the problem of lack of sub-grid small-scale structure information in variational optical flow estimation, we combine the motion laws of incompressible fluids. Introducing the idea of large eddy simulation, the instantaneous motion can be decomposed into large-scale motion and a small-scale turbulence in the data term. The Smagorinsky model is used to model and solve the small-scale turbulence. The improved subgrid scale Horn–Schunck (SGS-HS) optical flow algorithm provides better results in velocity field estimation of turbulent image sequences than the traditional Farneback dense optical flow algorithm. To make the SGS-HS algorithm equally competent for the open channel flow measurement task, a velocity gradient constraint is chosen for the canonical term of the model, which is used to improve the accuracy of the SGS-HS algorithm in velocimetric experiments in the case of the relatively uniform flow direction of the open channel flow field. The experimental results show that our algorithm has better performance in open channel velocimetry compared with the conventional algorithm.

## 1. Introduction

In recent years, the non-contact image sensor measurement scheme has been proposed to replace the traditional contact sensor measurement scheme in hydrological measurement tasks. This scheme uses an image velocity measurement algorithm to estimate the surface velocity of an open channel by processing the information obtained from image sensors. Computer vision projects have paid a great deal of attention to the research on fluid flow velocity estimation in image sequences. One of the main goals of this research is to retrieve precise velocity fields from image sequences. Video images of river, ocean, or weather flows provided by lab or field devices can help researchers in related fields gain a deeper understanding of complicated and unstable fluid motions for engineering tasks involving forecasting, monitoring, and prevention of related natural resources. Among the video flow measurement techniques, Particle Image Velocimetry (PIV) [1] is a well-established method and commonly used for flow visualization tasks. In [2], the authors proposed the Large-scale Particle Image Velocimetry (LSPIV) algorithm based on this technique [2], which uses natural tracers such as floating objects or ripples on the river surface to replace the particle tracers in the PIV technique. This approach improves the observation of the river flow field by the image algorithm and avoids contamination of the flow field by artificial tracers. However, the algorithm is less stable, and the natural tracers are highly susceptible to the influence of the surrounding environment on the river surface. Meanwhile, the LSPIV algorithm is based on correlation matching in the null domain for motion estimation, and its computational complexity is $O(n^{4})$, which is less efficient and requires more storage space. Fujita further proposed the Spatio-Temporal Image Velocimetry [3] (STIV)-based flow measurement algorithm in 2007. The efficient and stable characteristics of this algorithm have attracted a wide range of research, leading to its being extended and deepened. For example, Han et al. developed 2D spatio-temporal image velocimetry of a mountain river surface flow field based on UAV video [4], and Hamish et al. developed a novel airborne tracer particle distribution system [5].

Though all of the above methods have the general ability to describe complex fluid flows, none consider the incompressible fluid motion law when applied to river velocimetry. Through experimental studies, it has been found that optical flow algorithms are effective for fluid motion estimation. The dense optical flow method, such as the Farneback dense optical flow proposed by Farneback in 2003 [6], is more advantageous in global estimation, while the variational optical flow model proposed by Horn and Schunck [7] is well suited to incorporating constraints with physical interpretations in the construction of its data terms and regular terms, allowing the algorithm itself to optimize the optical flow field in conjunction with the hydrodynamics. Such combined hydrodynamic optical flow algorithms are gradually becoming a direction of exploration for many researchers.

The variational optical flow method proposed by Horn and Schunck [7] is itself a generalized energy minimization problem, with an objective function containing a data term and a canonical term. The data term is based on the Brightness Constancy, i.e., the luminance values of the same pixel point within the two preceding and following frames are conserved:(1)$$\frac{dI}{\;dt}=\frac{\partial I}{\partial t}+\nabla I\cdot u=0$$
where u=(u,v) is the velocity vector and *I* is the image grayscale value. The regular term is added to solve the aperture problem when estimating the velocity vector due to the lack of constraints related to the scalar equation in the data term. Based on the regular term with first-order constraints on the velocity gradient, the objective function for energy minimization is obtained by combining the data term
(2)$$J=\int_{\Omega }\left({\left(\frac{\partial I}{\partial t}+\nabla I\cdot u\right)}^{2}+\lambda \left({\left|\nabla u\right|}^{2}+{\left|\nabla v\right|}^{2}\right)\right)dx$$
where Ω is the image field, x=(x,y) is the pixel size, and the weighting factor λ is the weight of the two items before and after balancing in the function. Compared with the PIV and STIV methods, the variational optical flow method is able to estimate the dense velocity field and has advantages for obtaining global flow information. To better describe the fluid motion, Corpetti et al. [8] proposed an objective function based on the integral continuity equation (ICE) and second-order div-curl regularization, which can better recover the dispersion and vorticity of the fluid. Later, Liu et al. [9] discussed the relationship between the optical flow and the fluid in detail, and proposed using the equation of motion of the projection of the 3D object space on the 2D image plane as an optical flow constraint, which further provides an objective function with a physical interpretation. Among the techniques proposed in recent years, optical flow algorithms have been combined with methods such as orthogonal decomposition [10], wavelet expansions with higher-order regularization terms [11], optimal control schemes [12], and Bayesian stochastic filtering [13].

However, all of the above algorithms are based on the estimation of motion at the grid scale. For complex fluid motion, the subgrid small-scale eddy structure information cannot be ignored. In turbulent motion, energy is transferred and diffused between eddy structures of different scales. Generally, in traditional optical flow there are difficulties in accounting for turbulent motions at scales below the grid scale. Cassisa et al. [14] improved the accuracy of the optical flow algorithm for complex flows by introducing an eddy diffusion model with a subgrid transport equation instead of an optical flow constraint. The accuracy of the optical flow algorithm for estimating complex flows is improved by replacing the optical flow constraint with a subgrid transport equation; however, the diffusion coefficients of this transport equation are fixed and depend on empirical choices. Chen et al. [15] proposed the use of a structural subgrid model with vortex viscosity to calculate small-scale diffusion terms, providing an alternative solution for small-scale terms of the motion. Cai et al. [16] proposed a new strategy to estimate turbulent motion by first decomposing the Eulerian velocity of the fluid motion into a large-scale component and a small-scale turbulent component called position uncertainty, which they based on the derivation of Mémin [17], then combining the stochastic expression of the Reynolds transport theorem in order to derive the stochastic optical flow constraint equation. These works show that combining the turbulence model and variable fractional optical flow method is a meaningful research direction.

In this paper, we propose a new optical flow algorithm based on a sub-grid scale. This algorithm is used to solve the problem of the missing small-scale structure information of complex fluids in traditional optical flow. We first introduce large eddy simulation, then decompose the small-scale turbulence term from the instantaneous flow, and model its solution before introducing the velocity constraint derived from the energy conservation condition in the regular term with a certain penalty on the divergence and curl. In this way, the objective function has a data term containing the small-scale effect of the turbulence and a regular term scheme with a physical interpretation.

## 2. Method

### 2.1. Scalar Transmission Equation

We chose to use the scalar transport equation derived from the convective diffusion equation [18] to replace the Brightness Constancy. The scalar transport equation is used to describe the velocity field at a scalar concentration *C* in a fluid in a dimensionless form, as follows:(3)$$\frac{\partial C}{\partial t}+\nabla \cdot \left(Cu\right)-\frac{1}{ReSc}\Delta C=0$$
where $\nabla =\left(\frac{\partial }{\partial x},\frac{\partial }{\partial y}\right)$ is the Del operator, $\Delta =\frac{\partial^{2}}{\partial x^{2}}+\frac{\partial^{2}}{\partial y^{2}}$ is the Laplace operator, and Re and Sc are the Reynolds number and Schmidt number, respectively.

For a two-dimensional incompressible fluid with constant density and zero velocity divergence,
(4)∇·u=0.

Then, Equation (Equation 3) can be written as
(5)$$\frac{\partial C}{\partial t}+u\cdot \nabla C-\frac{1}{ReSc}\Delta C=0$$

In scalar images, the grayscale image *I* is related to a scalar concentration *C*, I∝∫Cdz, as proposed by Corpetti et al. [19], through the improvement of cloud motion estimation, in which *z* denotes the observation depth and is constant in two-dimensional images; as such, I∝C, and the grayscale image is proportional to the scalar concentration, allowing Equation (Equation 5) to be rewritten as
(6)$$\frac{\partial I}{\partial t}+u\cdot \nabla I-\frac{1}{ReSc}\Delta I=0$$

### 2.2. Data Term

The data term in the classical variational optical flow method is based on the Brightness Constancy proposed by Horn and Schunck [7]; however, the Brightness Constancy is relatively reliable only in the case of flow states with Schmidt number Sc≫1. Two inertial ranges exist for the scalar spectrum in the general scenario: the inertial convection range $\left(k<k_{\theta }\right)$, where the scalar variation is determined by the velocity, and the inertial diffusion range $\left(k_{\theta }<k<k_{\eta }\right)$, where the scalar variation is mainly influenced by the molecular diffusivity. To obtain the total velocity field in the general scenario, the scalar concentration field must contain information at least up to the Kolmogorov scale $k_{\theta }$ (Kolmogorov [20]).

In the optical flow algorithm, the information at the inertial diffusion range is lost due to the limitations on the spatial and temporal resolution of the acquired image. The grayscale image contains only the scalar concentration at pixel scale *C*, which corresponds to the range of large-scale motion estimation, i.e., the inertial convection range. For complex fluid motion, especially turbulent motion, the small-scale (sub-pixel scale) motion estimates (which are generally neglected during image acquisition) have a contribution to the flow field as well. In terms of physical structure, turbulence is formed by the superposition of various vortices with rotating structures at different scales, which are random in terms of their size and the directional distribution of their rotation axes; energy transfers can occur between them, and interactions between vortices at different scales cannot be neglected. In this work, we introduce the large eddy simulation theory [20], which decomposes the instantaneous flow into large-scale motion larger than the filter scale and small-scale motion smaller than the filter scale. In this approach, the grid scale is generally chosen as the filter scale. The grid size is the pixel size in the image sequence. From this, Equation (Equation 1) can be rewritten as follows:(7)$$\frac{\partial \bar{C}}{\partial t}+\nabla \cdot \left(\bar{uC}\right)-\frac{1}{ReSc}\Delta \bar{C}=0$$
where $\bar{C}$ is the resolvable large-scale scalar concentration. Similar to Large Eddy Simulation (LES) [21], the filtered nonlinear term in Equation (Equation 5) $\bar{uC}$ is here divided into the resolvable scale component and the residual component
(8)$$\left(\bar{uC}\right)=\bar{C}\bar{u}+\tau$$
where τ is the residual stress tensor. Then Equation (Equation 5) can be further written as
(9)$$\frac{\partial \bar{C}}{\partial t}+\nabla \cdot \left(\bar{C}\bar{u}\right)+\nabla \cdot \tau -\frac{1}{ReSc}\Delta \bar{C}=0$$

In the calculation of the subgrid residual term τ, Cassisa et al. [14] used a solution scheme related to the turbulent diffusion coefficient $D_{t}$, as follows:(10)$$\tau =-D_{t}\nabla \bar{C}$$
where the turbulent diffusion coefficient $D_{t}$ is defined as a statistical constant on the spatial domain [22]
(11)$$D_{t}=\eta {\left(\frac{3}{2C_{sgs}}\int_{K_{c}}^{\infty }E_{uu}\left(K\right)dK\right)}^{\frac{1}{2}}$$
where η is the grid size, the cutoff wave number $K_{c}=\frac{\pi }{\eta }$, and the constant $C_{sgs}$ is usually taken to be around 0.1 [22]. However, this modeling with a sub-pixel scale cannot obtain an explicit power spectrum when the velocity field is unknown $E_{uu}$, and the value of $D_{t}$ ultimately relies on empirical selection. Therefore, the solution scheme associated with the subgrid eddy viscosity $\mu_{t}$ [23] is chosen in this paper; the subgrid residual term can be obtained from the definition of generalized law of Newton inner friction
(12)$$\tau =2\rho \mu_{t}{\hat{S}}_{ij}$$
where ρ is the fluid density and ${\hat{S}}_{ij}$ is the strain rate tensor. The Smagorinsky model of large eddy simulation [23] is chosen in this paper to determine the subgrid eddy viscosity $\mu_{t}$, which is solved as follows:(13)$$\mu_{t}=\rho {\left(c_{s}\eta \right)}^{2}\left|{\hat{S}}_{ij}\right|$$
where ρ is the fluid density, the generic constant $c_{s}$ is generally taken to be between 0.1 and 0.2, and the strain rate tensor ${\hat{S}}_{ij}$ is calculated as follows:(14)$${\hat{S}}_{ij}=\frac{1}{2}\left(\frac{\partial \bar{u}}{\partial y}+\frac{\partial \bar{v}}{\partial x}\right)$$

A new data term based on the subgrid-scale variational optical flow model is thus constructed by adopting the scalar transport equation to replace the Brightness Constancy and by introducing large eddy simulation theory to decompose the large-scale term and the subgrid term. The new data term contains the contribution of small-scale motion, and is more consistent with the characteristics of complex fluid motion. The new data term are rewritten as follows:(15)$$J_{data}=\int_{\Omega }\left(\frac{\partial I}{\partial t}+u\cdot \nabla I+\nabla \cdot \tau -\frac{1}{ReSc}\Delta I\right)dx$$

### 2.3. Regular Term

In order to overcome the aperture problem, the variational optical flow requires the addition of a regular term to the objective function. While it is difficult to obtain good physical properties of the general regular term, in order to recover a degree of scattering and spin structure, Corpetti et al. [1] were able to use a second-order divergence-curl (div-curl) constraint instead of a first-order constraint on the velocity gradient, as follows:(16)$$\int_{\Omega }{∥\nabla \mathrm{div}u∥}^{2}+{∥\nabla \mathrm{curl}u∥}^{2}\;dx$$

However, this approach is computationally expensive and difficult to implement because its equation contains higher-order differentiation and penalizes vorticity too much. Because of this, the critical vorticity information is missing, causing errors in estimating the velocity. In contrast, the idea of Paul et al. [24] is to penalize the vorticity in the canonical term, with the Stokes equations as the constraint. The incompressible vorticity canonical term is derived from the Navier-Stokes equation as
(17)$$\int_{\Omega }\left(\alpha {\left(\omega -\omega_{T}\right)}^{2}+\beta {\left(\nabla \omega \right)}^{2}\right)dx$$
where ω=∇×u is the vorticity. Paul formulates a minimization problem based on the Horn–Schunck generalized function and determines the optimal source term and the boundary velocity for the Stokes equations under incompressible conditions to fully recover the velocity field. Although this method is suitable for fluids with less information on fluid elements, it restricts the class of incompressible fluids to Stokes-only fluids. Zille [25] and Chandrashekar [26] propose regularization schemes for incompressible conditions. In this work, the theory of large eddy simulation is introduced in the data term to decompose the sub-pixel scale term; however, in the actual application of the open channel velocimetry scenario, more small-scale eddies are essentially transmitted in the image with large-scale quasi-translational motion. Thus, we choose here to use the first order gradient regularization term, which penalizes both the divergence and curl of the flow field to some extent and add a more concise regularization term provided by the incompressible condition constraint:(18)$$\begin{array}{c} \int_{\Omega }{\left|\nabla u\right|}^{2}+{\left|\nabla v\right|}^{2}\;dx\\ \nabla \cdot u=0 \end{array}$$

The stochastic flow transport equation derived from Memin [17] from the perspective of energy conservation is taken as a regular term:(19)$$J_{regu}=\frac{1}{2}\int_{\Omega }\alpha {∥\nabla u∥}_{F}^{2}\;dx=\frac{1}{2}\int_{\Omega }\alpha \left({\left|\nabla u\right|}^{2}+{\left|\nabla v\right|}^{2}\right)dx$$
where ${|\nabla u∥}_{F}^{2}$ denotes the Frobenius parametrization of the representation matrix of the velocity vector, which is essentially the same as the canonical term of the classical optical flow; it follows that the classical velocity smoothing regularization can be interpreted as an energy conservation constraint for a uniform dispersion-free flow field.

In summary, the new objective function is established after combining the data term and the regular term to obtain the energy generalization function, as follows:(20)$$J=\int_{\Omega }\left({\left(\frac{\partial I}{\partial t}+u\cdot \nabla I+\nabla \cdot \tau -\frac{1}{ReSc}\Delta I\right)}^{2}+\lambda \left({\left|\nabla u\right|}^{2}+{\left|\nabla v\right|}^{2}\right)\right)dx$$

### 2.4. Discretization

Here, we denote $I_{t}=\frac{dI}{dt}$ and $\nabla I=\left(\frac{\partial I}{\partial x},\frac{\partial I}{\partial y}\right)=\left(I_{x},I_{y}\right)$. According to the Euler–Lagrange equation,
(21)$$\left\{\begin{array}{c} \frac{\partial J}{\partial u}-\frac{\partial }{\partial x}\frac{\partial J}{\partial u_{x}}-\frac{\partial }{\partial y}\frac{\partial J}{\partial u_{y}}=0\\ \frac{\partial J}{\partial v}-\frac{\partial }{\partial x}\frac{\partial J}{\partial v_{x}}-\frac{\partial }{\partial y}\frac{\partial J}{\partial v_{y}}=0 \end{array}\right.$$

The subpixel term τ can be considered homogeneous at the pixel scale, and its dependence on the local velocity *u* and its gradients $u_{x}$ and $u_{y}$ is weak. Therefore, it can be assumed that the partial derivatives of the local velocity and its gradient on the subpixel term τ are zero, and i=x,y is provided as
(22)$$\begin{array}{cc} \; & \frac{\partial \tau }{\partial u}=\frac{\partial \tau }{\partial v}=0\\ \; & \frac{\partial \tau }{\partial u_{i}}=\frac{\partial \tau }{\partial v_{i}}=0 \end{array}$$

Therefore, by bringing Equation (Equation 22) into Equation (Equation 21), simplification from Equation (Equation 23) yields
(23)$$\left\{\begin{array}{c} 2\left(I_{x}u+I_{y}v+I_{t}+\nabla \cdot \tau -\frac{1}{ReSc}\Delta I\right)I_{x}-2\cdot \lambda \Delta u=0\\ 2\left(I_{x}u+I_{y}v+I_{t}+\nabla \cdot \tau -\frac{1}{ReSc}\Delta I\right)I_{y}-2\cdot \lambda \Delta v=0 \end{array}\right.$$
where Δu can be discretely approximated as
(24)$$\begin{array}{c} \Delta u_{i,j}=\left(u_{i+1,j}-u_{i,j}\right)-\left(u_{i,j}-u_{i-1,j}\right)+\left(u_{i,j+1}-u_{i,j}\right)-\left(u_{i,j}-u_{i,j-1}\right)\\ =\left(u_{i+1,j}+u_{i-1,j}+u_{i,j+1}+u_{i,j-1}\right)-4u_{i,j} \end{array}$$

This results in $\Delta u=\kappa (\bar{u}-u)$, where $\bar{u}$ is the local average of *u* and κ takes the value determined by the discrete format, which is brought into Equation (Equation 24):(25)$$\left\{\begin{array}{c} \left(I_{x}^{2}+\lambda \right)u^{n+1}+I_{x}I_{y}v^{n+1}=\lambda {\bar{u}}^{n}-I_{x}\nabla \cdot \tau +\frac{1}{ReSc}I_{x}\Delta I-I_{x}I_{t}\\ I_{x}I_{y}u^{n+1}+\left(I_{y}^{2}+\lambda \right)v^{n+1}=\lambda {\bar{v}}^{n}-I_{y}\nabla \cdot \tau +\frac{1}{ReSc}I_{y}\Delta I-I_{y}I_{t} \end{array}\right.$$

The iterative format of the velocity vector u can be obtained by deforming and simplifying Equation (Equation 25) as follows:(26)$$\left\{\begin{array}{c} u^{n+1}={\bar{u}}^{n}-\frac{I_{x}{\bar{u}}^{n}+I_{y}{\bar{v}}^{n}+I_{t}+\nabla \cdot \tau -\frac{1}{ReSc}\Delta I}{\lambda +I_{x}^{2}+I_{y}^{2}}I_{x}\\ v^{n+1}={\bar{v}}^{n}-\frac{I_{x}{\bar{u}}^{n}+I_{y}{\bar{v}}^{n}+I_{t}+\nabla \cdot \tau -\frac{1}{ReSc}\Delta I}{\lambda +I_{x}^{2}+I_{y}^{2}}I_{y} \end{array}\right.$$

Although variable spectral flow is a computer vision method with good tracking and reproduction for small-scale displacements, its derivation about time and the spatial gradient does not satisfy the approximation relationship when there are large-scale displacements between the front and back images. To solve this problem, the multi-scale image pyramid algorithm is usually used. The Algorithm 1 itself is structured by a series of low-pass filters that allow the image to successively reduce its resolution by quadratic sampling, followed by solving the updated velocity field from coarse to fine, as in Figure 1.
**Algorithm 1** Multiresolution Pyramid SGS-HS Algorithm**Input:** Image Sequence**Output:** Optical Flow Velocity Field
 1:Read two images before and after Img1 and Img2 2:Image preprocessing (perspective transformation, etc.) 3:Image pyramid downsampling (L Layers) 4:**for** int l = L to 0 **do** 5:    **if** condition **then** 6:        Initialize the velocity field 7:    **else** 8:        Interpolating the upper-velocity field to the next 9:    **for** Image Layer Update **do** 10:        Optimal Solution 11:        Computing image gradients and Laplacian operators 12:        Calculating the velocity field from the model of Equation (Equation 26) 13:        Update velocity field 14:**return** result


## 3. Experiment and Analysis

### 3.1. Scalar Image Sequences

The image sequence is taken from the dataset of the French team Cemagref. Each image has a pixel resolution of 256×256, and the turbulent flow represented in the image is isotropic and homogeneous, with Reynolds number Re=3000 and Schmidt number Sc=0.7. In the scalar image experiments, a velocity field comparison of conventional FB [6], TV-L1 [27], Fast optical flow using dense inverse search (DIS) [28], and SGS-HS is performed in this work, with the results shown in Figure 2.

As can be seen in Figure 2c–e, the velocity fields of the conventional FB, TV-L1, and DIS are very smooth and perform generally in describing complex fluid motion, as the core of its algorithm relies on the Brightness Constancy. Figure 2f shows the velocity field of the SGS-HS algorithm proposed in this paper after adding the multi-resolution pyramidal image algorithm. Compared with the other algorithms, the velocity field in Figure 2f is more refined and performs better for the vorticity of the flow field and the degree of bending of certain flows. The purpose of this experiment is to compare the ability of the algorithm to describe the flow field when processing synthetic turbulent images; as can be seen from the comparison of the velocity field results in Figure 2. SGS-HS algorithm performs better.

### 3.2. Turbulence Video

This experiment aims to test the ability of FB, TV-L1, DIS and SGS-HS to describe incompressible turbulent flow fields in nature. The experimental video resolution is 1920×1080; its cut-frame image and the HSV optical flow fields for all algorithms are shown in Figure 3.

It can be seen from the experimental results that FB, TV-L1, and DIS have a poor ability to describe the complex flow in the face of such natural incompressible turbulent flow as in Figure 3a,b. Moreover, it can be seen that in the HSV optical flow diagram in Figure 3c the overall flow performance is poor, and cannot capture the turbulent flow. The velocity field flow description in Figure 3d,e is more ambiguous, and the velocity gradient is too smooth. In contrast, SGS-HS can capture the flow direction and flow state of the flow field better in the turbulent video experiment because of the inclusion of the subgrid small-scale term of the large eddy simulation theory, as shown in the HSV optical flow map in Figure 3f. It can be seen that the overall flow state of the flow field is more clearly represented and the HSV optical flow map is highly consistent with the actual turbulent image when the local turbulent region is enlarged in Figure 4. This indicates that the SGS-HS algorithm can provide a better description of the turbulent flow field.

### 3.3. Open Channel Flow Measurement

This experiment evaluates the velocimetric capability of the SGS-HS in a real open channel measurement task. The measurement site is an open channel section of a hydrological station; in hydrographic measurement experiments, the current meter is generally used as the agreed true value for comparison criteria. The model of rotor current meter we selected is the LS25-3A, and its number is 141249. In this experiment, LSPIV [2], STIV [3], FB [6], TV-L1 [27], DIS [28], and SGS-HS are selected as the measurement indexes for comparison experiments using the vertical average flow velocity, average flow velocity, and discharge.

The arrangement of ground calibration points of the open channel is shown in Figure 5. Two points, A and B, were arranged on the left bank, and two other points, D and C, were arranged on the right bank, with the starting point coinciding with point D and the ending point coinciding with point A. Line DA was used as the cross-section line. The flow velocity meter measurement results are shown in Table 1.

The video resolution of the open channel experiment was 1920×1080 and the duration was 10s, with a total of 325 image sequences of cut frames. Perspective transformation was used to convert the motion vector into the two-dimensional flow field from the image coordinate system to the world coordinate system in order to obtain the real displacement between two frames, from which the river surface velocity can be calculated. Then the average velocity and cross-sectional discharge can be obtained by the velocity area method [29], as in Equation (Equation 27):(27)$$\begin{array}{c} Q=\sum_{i=1}^{n}q_{i}=\sum_{i=1}^{n}v_{i}S_{i}\\ \bar{v}=\frac{Q}{S} \end{array}$$
where $v_{i}$ is the vertical average velocity, $s_{i}$ is the cross section, the discharge is $q_{i}=v_{i}s_{i}$, *Q* is the total discharge of the river, and $\bar{v}$ is the average velocity. LSPIV [2], STIV [3], FB [6], TV-L1 [27], DIS [28], and SGS-HS were selected for the comparison experiment, and the results are shown in Table 2, Table 3 and Table 4.

From Figure 6a, it can be seen that FB and SGS-HS are susceptible to the interference of the perspective transformation accuracy and the environment around the open channel in the velocimetry experiment, which is due to their own global dense optical flow estimation, meaning that the experimental results fluctuate more near the banks on both sides. TV-L1 and DIS have relatively stable performance in terms of their overall results. In the middle basin of the open channel, that is, from 5 m to 8 m, FB and SGS-HS have better performance. In particular, the velocity measurement results of SGS-HS have the best accuracy and the least fluctuation, as shown in Table 3 and Table 4.

For the calculation of the average velocity and discharge in the flow field, as shown in Figure 6b,c, the accuracy of the results of LSPIV, STIV, and TV-L1 is inferior to that of FB, DIS, and SGS-HS. From an analysis of Table 2, Table 3 and Table 4, it can be seen that SGS-HS has a certain accuracy in the overall flow field discharge and average velocity, although the average velocity of the vertical line at the starting distance on both sides of the open channel fluctuates widely.

The experimental equipment processor used for our experiments was an AMD Ryzen7 4800H, and the program running environment was Python3.9. A comparison of the time consumption of the four algorithms is shown in Table 5 and Figure 6d.

## 4. Conclusions

This work proposes the subgrid variational optimal optical flow estimation algorithm for measuring video flow. The Smagorinsky model of large eddy simulation is introduced to solve the small-scale turbulence terms decomposed by transient flow in the data items, meaning that the overall variational model in processing turbulent image tasks has better performance compared to the conventional dense optical flow algorithm. This solves the issue of missing subgrid small-scale structural information in variational optical flow estimation based on the grid scale. The SGS-HS algorithm takes into account the consistency of the river flow in the actual open channel velocimetry task and applies a velocity gradient constraint to the regular term, which is effectively a penalty for velocity divergence and curl. The results demonstrate that after correctly raising the canonical weights the SGS-HS algorithm can complete the task of open channel velocimetry with <1% relative error for the average velocity and flow rate.

Because the SGS-HS algorithm is itself an iterative solution for the global velocity, it has ordinary performance in real time and is sensitive to both light changes and surrounding environmental disturbances. The SGS-HS algorithm essentially relies on the motion provided by the fluidly stable brightness values in the image. However, illumination changes can interfere with the SGS algorithm’s estimation of fluid motion, and perspective transformation distort fluid motion to an extent in open channel tasks. Therefore, our results on both sides of the open channel have certain errors. Future work coud be directed towards improving the robustness of the algorithm to illumination changes and improving the optical flow network for deep learning by combining the variational optimization model. In this way, the accuracy and computational efficiency of the algorithm model could be further improved.

## Figures and Tables

**Figure 1 sensors-23-00437-f001:**
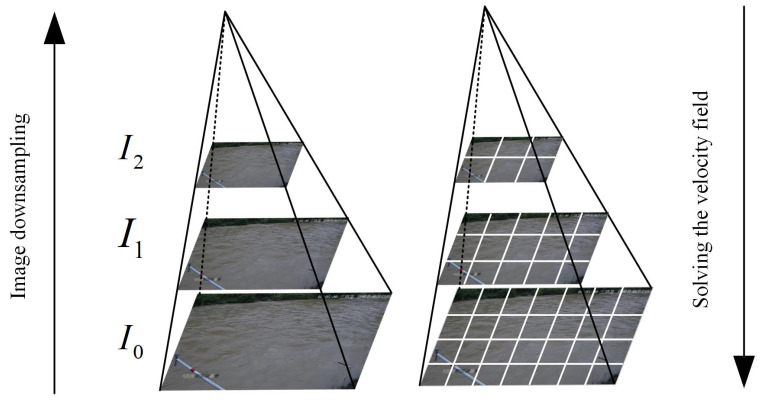
Image pyramid.

**Figure 2 sensors-23-00437-f002:**
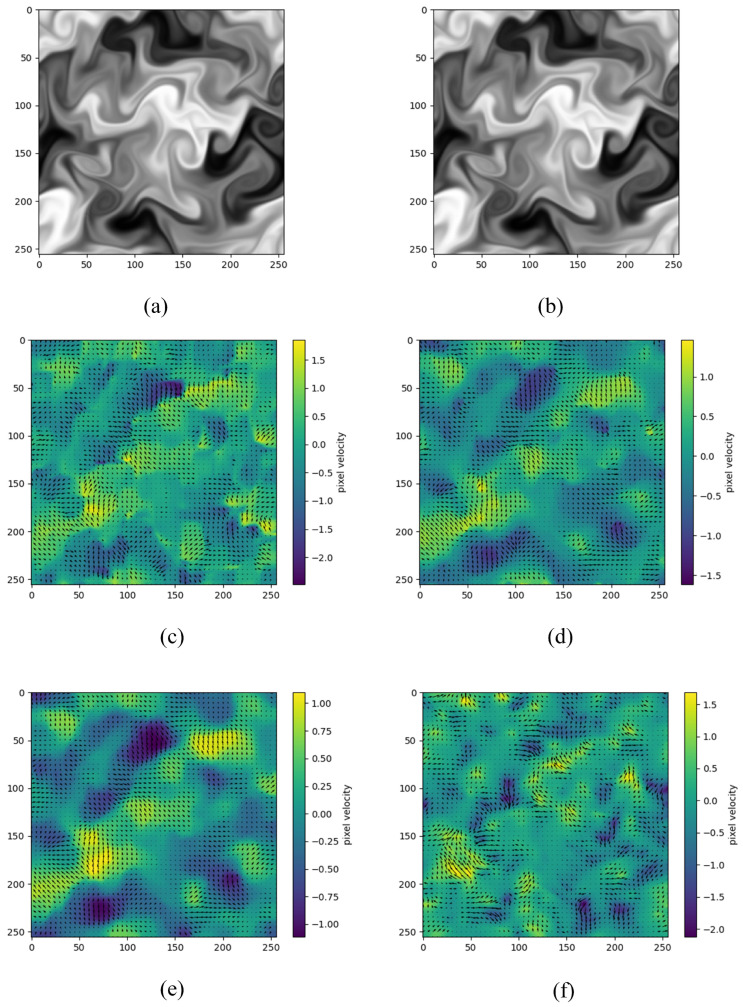
Scalar images and their velocity fields: (**a**,**b**) scalar image adjacent to two frames; (**c**) FB (t=60); (**d**) TV-L1 (t=60); (**e**) DIS (t=60); (**f**) SGS-HS (t=60).

**Figure 3 sensors-23-00437-f003:**
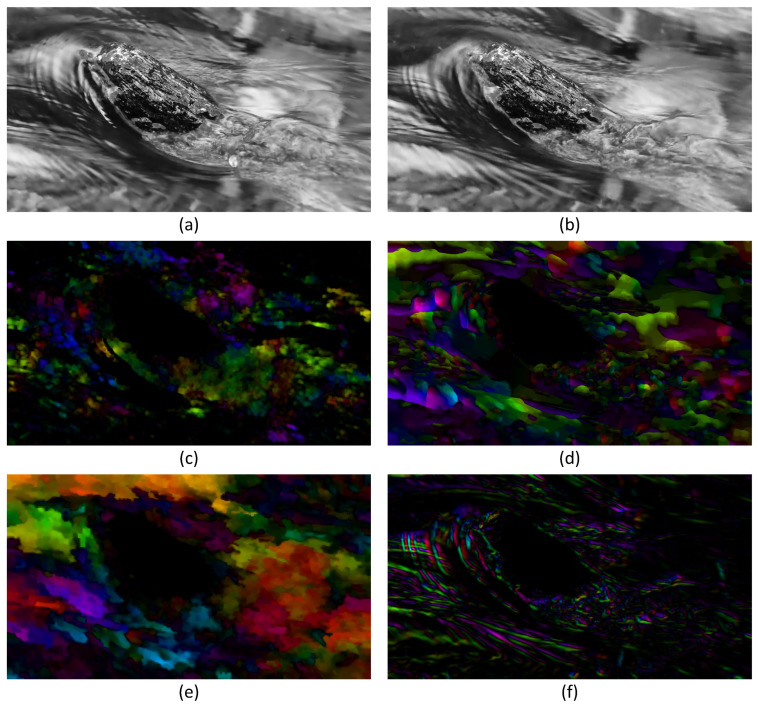
Experimental plots of turbulence video: (**a**,**b**) two adjacent frames of turbulence video; (**c**) HSV optical flow map of FB algorithm; (**d**) HSV optical flow map of TV-L1; (**e**) HSV optical flow map of DIS; (**f**) HSV optical flow map of SGS-HS.

**Figure 4 sensors-23-00437-f004:**
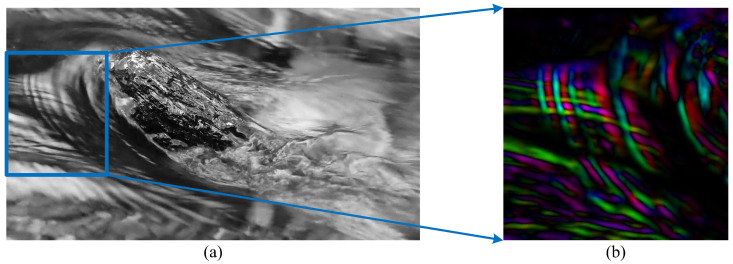
Turbulence image and its locally scaled HSV optical flow diagram: (**a**) turbulence image and (**b**) locally scaled HSV optical flow diagram of turbulence image.

**Figure 5 sensors-23-00437-f005:**
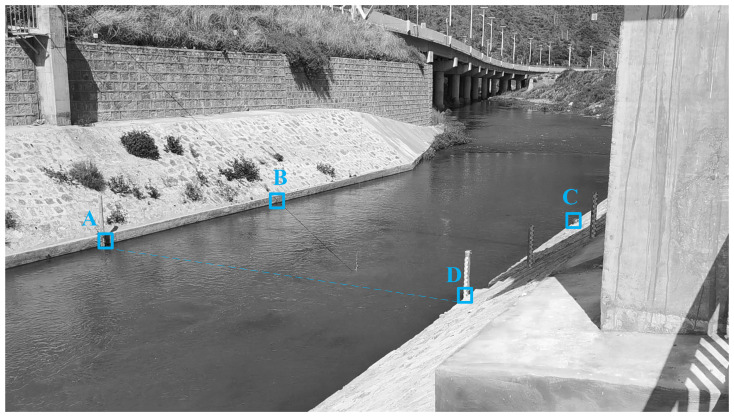
Open channel ground marking point.

**Figure 6 sensors-23-00437-f006:**
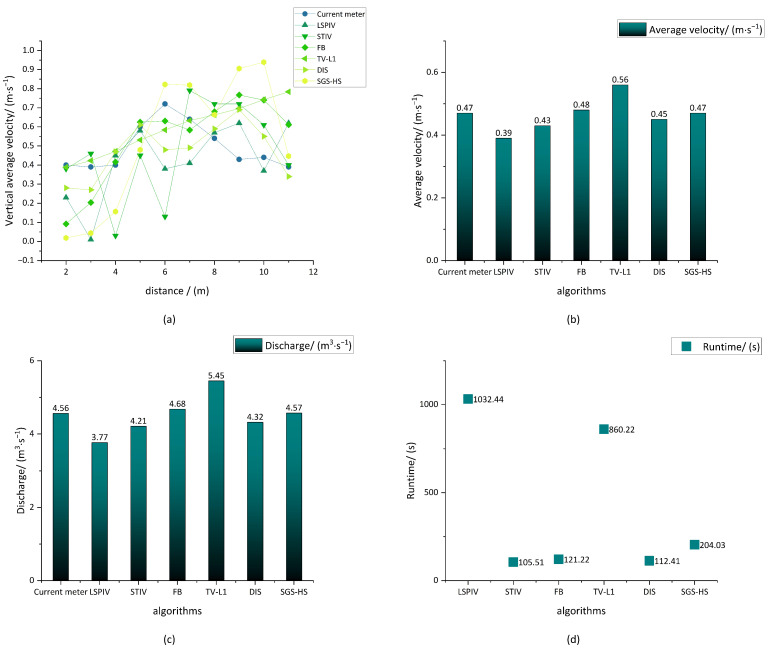
Comparison of experimental results: (**a**) vertical average velocity; (**b**) average velocity; (**c**) discharge; (**d**) time consumption.

**Table 1 sensors-23-00437-t001:** Current meter measurement results.

Distance/(m)	Vertical Average Velocity/($m\cdot s^{-1}$)	Segment Mean Velocity/($m\cdot s^{-1}$)	Segment Area/($m^{2}$)	Segment Discharge /($m^{3}\cdot s^{-1}$)
0 (side)	0			
0∼2		0.29	1.45	0.42
	0.36			
2∼3		0.4	0.85	0.34
	0.43			
3∼4		0.39	0.92	0.36
	0.35			
4∼5		0.4	0.99	0.4
	0.45			
5∼6		0.6	0.98	0.59
	0.76			
6∼7		0.72	0.95	0.68
	0.68			
7∼8		0.64	0.8	0.51
	0.61			
8∼9		0.54	0.77	0.42
	0.48			
9∼10		0.43	0.86	0.37
	0.38			
10∼11		0.44	0.71	0.31
	0.49			
11∼11.9		0.39	0.42	0.16
11.9 (side)	0			

**Table 2 sensors-23-00437-t002:** Comparison of the results of the six algorithms and the current meter.

Vertical Average Velocity/($m\cdot s^{-1}$)											Average Velocity /($m\cdot s^{-1}$)	Discharge /($m^{3}\cdot s^{-1}$)
Distance/(m)	2	3	4	5	6	7	8	9	10	11		
Current meter	0.36	0.43	0.35	0.45	0.76	0.68	0.61	0.48	0.38	0.49	0.47	4.56
LSPIV	0.23	0.01	0.45	0.58	0.38	0.41	0.57	0.62	0.37	0.62	0.39	3.77
STIV	0.38	0.46	0.031	0.45	0.13	0.79	0.72	0.72	0.61	0.4	0.43	4.21
FB	0.092	0.2	0.41	0.62	0.63	0.58	0.67	0.76	0.73	0.61	0.48	4.68
TV-L1	0.39	0.42	0.47	0.53	0.58	0.63	0.66	0.69	0.74	0.78	0.56	5.45
DIS	0.28	0.27	0.47	0.61	0.48	0.49	0.59	0.69	0.55	0.34	0.45	4.32
SGS-HS	0.018	0.044	0.16	0.48	0.82	0.81	0.66	0.9	0.93	0.44	0.47	4.57

**Table 3 sensors-23-00437-t003:** Absolute errors of the velocity measurement results of the six algorithms.

Absolute Error/($m\cdot s^{-1}$)											Average Velocity /($m\cdot s^{-1}$)	Discharge /($m^{3}\cdot s^{-1}$)
Distance/(m)	2	3	4	5	6	7	8	9	10	11		
LSPIV	0.13	0.42	0.1	0.13	0.38	0.27	0.04	0.14	0.01	0.13	0.08	0.79
STIV	0.02	0.03	0.32	0	0.63	0.11	0.11	0.24	0.23	0.09	0.06	0.35
FB	0.26	0.23	0.06	0.17	0.13	0.1	0.06	0.28	0.35	0.12	0.01	0.12
TV-L1	0.03	0.006	0.12	0.08	0.17	0.05	0.05	0.21	0.36	0.29	0.09	0.89
DIS	0.08	0.16	0.12	0.16	0.28	0.19	0.02	0.21	0.17	0.15	0.02	0.24
SGS-HS	0.34	0.38	0.19	0.03	0.06	0.13	0.05	0.42	0.55	0.05	0.003	0.01

**Table 4 sensors-23-00437-t004:** Relative errors of the velocity measurement results of the six algorithms.

Relative Error/(%)											Average Velocity /($m\cdot s^{-1}$)	Discharge /($m^{3}\cdot s^{-1}$)
Distance/(m)	2	3	4	5	6	7	8	9	10	11		
LSPIV	36.11	97.7	28.57	28.89	50.21	40.15	7.01	28.28	3.61	26.86	17.02	17.32
STIV	5.56	6.98	91.43	0.8	82.81	16.82	18.01	49.63	61.51	17.73	12.76	7.68
FB	72.22	53.49	17.14	37.78	17.11	14.71	9.83	58.33	92.1	24.49	2.12	2.63
TV-L1	7.94	1.53	34.89	18.12	23.15	6.94	8.65	44.94	95.47	59.89	19.14	19.51
DIS	22.2	37.2	34.3	35.6	36.8	27.3	3.3	43.8	44.7	30.6	4.3	5.3
SGS-HS	94.44	88.37	54.28	6.67	7.89	19.12	8.2	87.5	144.7	10.2	0.64	0.22

**Table 5 sensors-23-00437-t005:** Algorithm runtimes.

Runtime Environment	Runtime/(s)					
	LSPIV	STIV	FB	TV-L1	DIS	SGS-HS
AMD Ryzen 7	1032.44	105.51	121.22	860.22	112.41	204.03
Windows 10 (x64)						
Python 3.9						
OpenCV4.2						

## Data Availability

Not applicable.

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
