# Peer review of "Subgrid Variational Optimized Optical Flow Estimation Algorithm for Image Velocimetry"

_sensors, 2022, doi:10.3390/s23010437_

Round 1

Reviewer 1 Report

Dear Editor,

I find this article interesting and valuable as it improves the optical flow algorithm by using additional constraints obtained from mathematical theory of incompressible fluid flows, namely the Large Eddy Simulation for modelling turbulence. This research therefore can be interesting to both groups of researchers, computer vision community and fluid dynamics, turbulence modelling community.

However - I have some concerns regarding the clarity of writting and I suggest authors to carefully check the article one more time, sentence by sentence, to make sure that the intention and message are clear.

Here are a few comments:

Comment 1, abstract:
"To solve the problem of missing information on subgrid small-
scale structure in the estimation of variational optical flow based on a grid-scale, combined with
the physical motion law of incompressible fluid, we introduce the idea of large eddy simulation,
decomposes the instantaneous motion into a large-scale motion term and a small-scale turbulence
term in the variational model data item, and uses Smagorinsky model to solve the small-scale
turbulence term."

Advice:
Please reformulate the sentence, maybe brake it down into shorter ones and make it clear what is happening. I understand what is the goal because I understand the LES but this is not clear enough for academic paper. Make it clear what is the role of Large Eddy Simulation in your study. We decompose motion into large scale motion and sub-grid scale motion within the additive decomposition, but you jump into discussion using 'terms', then you use very generic "variational model data item", all this makes a very confusing sentence.

Comment 2, Lines 69+
"In the scenario of complex fluid motion, the subgrid small-scale vortex structure
is not negligible for the fluid, and the energy is transferred and diffused between vortex
structures at different scales in turbulent flow, which cannot be reflected at the grid scale"

So you write: 1) "energy is transferred and diffused between vortex
structures at different scales in turbulent flow". Yes this is true.
2) "which cannot be reflected at the grid scale" But, it can be for the for the scales above the grid scale.

Advice: Please reformulate the sentence - make sure it is pointed out to the reader that we have difficulties with accounting for the turbulent motions at scales below the grid scale.

Comment 3, line 72+:
"Cassisa et al.[14 ] improved the accuracy of the optical flow algorithm for complex flows
by introducing an eddy diffusion model ..."

Comment: How is the model you use different from this one? The Smagorinsky model also represents eddy diffusion. Also note, nowhere later have the grid scale have been clearly i.e. explicitly defined.

Comment 4, line 86+
"In this paper, we propose the Subgrid Scale Horn-Schunck(SGS-HS) optical flow
algorithm to solve the problem of missing information on the subgrid’s small-scale structure
based on the subgrid estimation optical flow at the grid scale."

Lets have a close-up:
"subgrid estimation optical flow at the grid scale". I am not sure what this is?

Comment 5, page 4 at the bottom:
"subgrid vortex viscosity" - it is usually "eddy viscosity", sometimes "turbulence viscosity" never vortex viscosity.

Please revise the article so the intentions of the authors are clear. This is a case of academic writing that needs improvement not English language.

I recommend revision on these lines after which the paper can be accepted for publication if everything is done properly.

Author Response

Dear Reviewer:

We quite appreciate for your favorite consideration and the insightful comments. Now we have revised the manuscript (ID: sensors-2088656) exactly according to your comments, and found these comments are very helpful. Revised portion are marked in blue in the manuscript. The revisions were addressed point by point below.

Comment 1:

Please reformulate the sentence, maybe brake it down into shorter ones and make it clear what is happening. I understand what is the goal because I understand the LES but this is not clear enough for academic paper. Make it clear what is the role of Large Eddy Simulation in your study. We decompose motion into large scale motion and sub-grid scale motion within the additive decomposition, but you jump into discussion using 'terms', then you use very generic "variational model data item", all this makes a very confusing sentence.

Response:

We have reformulated the sentence according to the Reviewer’s comment.

“To solve the problem of lack of sub-grid small-scale structure information in variational optical flow estimation, we combine the motion laws of incompressible fluid. Introducing the idea of large eddy simulation, the instantaneous motion can be decomposed into a large-scale motion and a small-scale turbulence in the data term. The Smagorinsky model is used to model and solve the small-scale turbulence.”

Comment 2:

Please reformulate the sentence - make sure it is pointed out to the reader that we have difficulties with accounting for the turbulent motions at scales below the grid scale.

Response:

We have reformulated the sentence according to the Reviewer’s comment.

“For complex fluid motion, the sub-grid small-scale eddy structure information cannot be ignored. In turbulent motion, energy is transferred and diffused between eddy structures of different scales. Generally, in traditional optical flow, we have difficulties with accounting for the turbulent motions at scales below the grid scale.”

Comment 3:

How is the model you use different from this one? The Smagorinsky model also represents eddy diffusion. Also note, nowhere later have the grid scale have been clearly i.e. explicitly defined.

Response:

Cassisa[14] model, the main disadvantage that the power spectrum Euu cannot be obtained explicitly if the velocity field is unknown. Thus, the value of DT (a turbulent diffusion coefficient) remains indeterminable. We propose to use the Smagorinsky model that is essentially an energy cascade from large scales to smaller ones. And the parameters of the model can be determinable.

We have added the definition of grid size in the manuscript.

“The grid size is the pixel size in the image sequence.”

Comment 4:

Lets have a close-up: "subgrid estimation optical flow at the grid scale". I am not sure what this is?

Response:

We are very sorry for our misexpression and have reformulated the sentence.

“In this paper, we propose the new optical flow algorithm based on sub-grid scale. The algorithm is used to solve the problem of missing small-scale structure information of complex fluids in traditional optical flow.”

Comment 5:

"subgrid vortex viscosity" - it is usually "eddy viscosity", sometimes "turbulence viscosity" never vortex viscosity. Please revise the article so the intentions of the authors are clear. This is a case of academic writing that needs improvement not English language.

Response:

We are very sorry for our incorrect writing and have changed “vortex viscosity” to “eddy viscosity”.

We have also improved the academic writing elsewhere in the paper.

In all, we found the reviewer’s comments are quite helpful, and we revised my paper point-by-point. Thank you and the review again for your help!

Reviewer 2 Report

Obviously, the original intention and starting point of this study are both good, but there are still some areas worth discussing in terms of specific research details. If the authors revise this manuscript, they are suggested to consider the following points:

-The authors mentioned "forecasting" in the abstract, but did not mention other relevant details about forecasting in the full text. Does this mean that the research results can be used in other relevant research on forecasting? Otherwise, the author is suggested to delete it, or the author can better explain it so that readers can understand this study.

-Although the accuracy of the method proposed by the authors in this study is improved to a certain extent compared with the traditional method, the variability of the comparison error between the method and the measured value at different distances is large? Does it indicate that the method has low robustness? It is necessary for the authors to explain in detail what causes the error variability of the methods proposed in this study. More importantly, most of the relative errors are greater than 5%, which is obviously not suitable for the error requirements of engineering applications. It is also suggested that the authors analyze the factors restricting engineering applications.

-It seems that this study only focuses on images, not images or even videos in the full sense. Can this study be extended to video analysis? What are the problems?

Author Response

Dear Reviewer:

We are very grateful to your comments for the manuscript (ID: sensors-2088656). According with your advice, we tried our best to amend the relevant part and made some changes in the manuscript. All of your questions were answered below. And here we list the changes and marked in orange in revised manuscript.

-The authors mentioned "forecasting" in the abstract, but did not mention other relevant details about forecasting in the full text. Does this mean that the research results can be used in other relevant research on forecasting? Otherwise, the author is suggested to delete it, or the author can better explain it so that readers can understand this study.

Response:

We are very sorry for our incorrect writing and have deleted “forecasting” in the abstract.

-Although the accuracy of the method proposed by the authors in this study is improved to a certain extent compared with the traditional method, the variability of the comparison error between the method and the measured value at different distances is large? Does it indicate that the method has low robustness? It is necessary for the authors to explain in detail what causes the error variability of the methods proposed in this study. More importantly, most of the relative errors are greater than 5%, which is obviously not suitable for the error requirements of engineering applications. It is also suggested that the authors analyze the factors restricting engineering applications.

Response:

It is really true as Reviewer suggested that the method has low robustness. We have explained in detail what causes the error variability of the method. The details are as follow:

“The SGS-HS algorithm essentially relies on the motion provided by the fluidly stable brightness values in the image. Illumination changes can interfere with the SGS algorithm's estimation of fluid motion. Perspective transformation also distort fluid motion to some extent in open channel tasks. Therefore, our results on both sides of the open channel will have certain errors.”

Regarding, our algorithm cannot provide more accurate results of vertical average velocity. We can give relatively accurate results in the average velocity and discharge of the cross-section that are usually required in engineering applications. Factors restricting engineering applications generally include the angle of camera installation, surrounding environment and Illumination changes.

-It seems that this study only focuses on images, not images or even videos in the full sense. Can this study be extended to video analysis? What are the problems?

Response:

This study can be extended to video analysis, but its key point is still to cut video frames into image sequences. The algorithm essentially performs optical flow estimation on two adjacent frames of images.

Once again, thank you very much for your comments and suggestions.

Reviewer 3 Report

In this paper, a subgrid scale optimization method is studied to simulate the complex fluid flow in image sequences and estimate its two-dimensional velocity field. Experiments are carried out on turbulence image sequences and open channel flow measurement tasks. However, the following aspects need to be improved:

(a)The difference between the subgrid method proposed by the author and Cassisa et al. [14] does not involve much.

(b)Turbulence contrast experiment can not well explain the advantages of the proposed method, lacking ground truth and numerical comparison information.

(c)The experimental comparison algorithm seems to be insufficient and outdated. As far as I know, there are many dense optical flow algorithms (including but not limited to deep learning, such as TV-L1, etc.).

Author Response

Dear Reviewer:

Thank you for your comments concerning our manuscript (ID: sensors-2088656). Those comments are all valuable and very helpful for revising and improving our paper. We have studied comments carefully and have made correction which we hope meet with approval. Revised portions are marked in green in the manuscript. The main corrections in the paper and the responds to the reviewer’s comments are as flowing:

(a) The difference between the subgrid method proposed by the author and Cassisa et al. [14] does not involve much.

Response:

Cassisa[14] model, the main disadvantage that the power spectrum Euu cannot be obtained explicitly if the velocity field is unknown. Thus, the value of DT (a turbulent diffusion coefficient) remains indeterminable. We propose to use the Smagorinsky model that is essentially an energy cascade from large scales to smaller ones. And the parameters of the model can be determinable.

(b) Turbulence contrast experiment can not well explain the advantages of the proposed method, lacking ground truth and numerical comparison information.

Response:

We are very sorry for lacking ground truth and numerical comparison information. We are currently unable to offer turbulent ground truth and can only be compared by visualizing the optical flow map.

(c) The experimental comparison algorithm seems to be insufficient and outdated. As far as I know, there are many dense optical flow algorithms (including but not limited to deep learning, such as TV-L1, etc.).

Response:

We have added TV-L1[27] and DIS[28] in each experiment and compared the algorithms. Figure 2, 3, 6 and Table 2, 3, 4, 5 have been revised in the manuscript.

[27] Zach C, Pock T, Bischof H. A duality based approach for realtime tv-l 1 optical flow[C]//Joint pattern recognition symposium. Springer, Berlin, Heidelberg, 2007: 214-223.

[28] Kroeger T, Timofte R, Dai D, et al. Fast optical flow using dense inverse search[C]//European conference on computer vision. Springer, Cham, 2016: 471-488.

Once again, thank you very much for your comments and suggestions.
